# Phenazine-1-carboxylic Acid Produced by *Pseudomonas* *chlororaphis* YL-1 Is Effective against *Acidovorax citrulli*

**DOI:** 10.3390/microorganisms9102012

**Published:** 2021-09-23

**Authors:** Youzhou Liu, Yaqiu Zhou, Junqing Qiao, Wenjie Yu, Xiayan Pan, Tingting Zhang, Yongfeng Liu, Shi-En Lu

**Affiliations:** 1Institute of Plant Protection, Jiangsu Academy of Agricultural Sciences, Nanjing 210014, China; shitouren88888@163.com (Y.L.); yaqiuzhou@163.com (Y.Z.); Junqingqiao@163.com (J.Q.); Xiayanpan20210821@163.com (X.P.); 2Department of Biochemistry, Molecular Biology, Entomology and Plant Pathology, Mississippi State University, Starkville, MS 39759, USA; 3College of Plant Protection, Nanjing Agricultural University, Nanjing 210095, China; wenjieyu20210817@163.com (W.Y.); tingtingzhang0817@163.com (T.Z.)

**Keywords:** *Acidovorax citrulli*, *Pseudomonas chlororaphis*, phenazine-1-carboxylic acid

## Abstract

The bacterial pathogen *Acidovorax citrulli* causes the destructive fruit blotch (BFB) on cucurbit plants. *Pseudomonas chlororaphis* YL-1 is a bacterial strain isolated from Mississippi soil and its genome harbors some antimicrobial-related gene clusters, such as phenazine, pyrrolnitrin, and pyoverdine. Here, we evaluated the antimicrobial activity of strain YL-1 as compared with its deficient mutants of antimicrobial-related genes, which were obtained using a *sacB-*based site-specific mutagenesis strategy. We found that only phenazine-deficient mutants Δ*phzE* and Δ*phzF* almost lost the inhibitory effects against *A. citrulli* in LB plates compared with the wild-type strain YL-1, and that the main antibacterial compound produced by strain YL-1 in LB medium was phenazine-1-carboxylic acid (PCA) based on the liquid chromatography-mass spectrometry (LC-MS) analysis. Gene expression analyses revealed that PCA enhanced the accumulation of reactive oxygen species (ROS) and increased the activity of catalase (CAT) in *A. citrulli*. The inhibition effect of PCA against *A. citrulli* was lowered by adding exogenous CAT. PCA significantly upregulated the transcript level of *katB* from 6 to 10 h, which encodes CAT that helps to protect the bacteria against oxidative stress. Collectively, the findings of this research suggest PCA is one of the key antimicrobial metabolites of bacterial strain YL-1, a promising biocontrol agent for disease management of BFB of cucurbit plants.

## 1. Introduction

Bacterial fruit blotch (BFB), caused by *Acidovorax citrulli*, is a destructive disease that infects cucurbit plants [1]. The disease was first found in Florida in the USA in 1989 [2]. In recent years, the disease has been involuntarily spread to many parts of the world by contaminated commercial seeds [1]. It was reported that seeds even with a low level of *A. citrulli* containment can result in severe BFB epidemics under a favorable environment [3]. Besides fruit blotch, *A. citrulli* also causes leaf blight, seedling blight, and/or blossom rot of cucurbitaceous plants. Annual yield loss of watermelon caused by BFB in the USA reached 5–50%, depending on the stage of infection and environmental conditions [4,5].

Since BFB is a seed-borne disease [6], control of BFB mainly depends on use of *A. citrulli*-free seeds and on treatment of *A. citrulli*-contaminated seeds either with physical measures such as dry heat or with antibacterial chemicals such as thiamphenicol [7,8]. However, the use of preventive control measures as described above has led to some negative impacts on seed quality, including reduction of seed germination and seedling growth [9]. Therefore, alternative safe measures for control of BFB are needed. It was reported that the decontamination method to inactivate *A. citrulli* on Cucurbitaceae seeds without loss of seed viability is to be sequentially treated by aqueous chlorine dioxide, drying, and dry heat [10].

Biological control is considered an ecologically sound and environmentally friendly approach for management of plant diseases [11]. Previous studies showed that selected biological control agents (BCAs) can be used to control BFB on cucurbit crops, including bacteriophage [12], nonpathogenic strains of *A. citrulli* and *A. avenae* [13], antagonistic bacteria such as *Bacillus* spp. [14,15], *Paenibacillus lentimorbus* [16], and *Pseudomonas*
*fluorescens* [17], antagonistic yeasts such as *Pichia anomala* [18], *Rhodotorula aurantiaca*, and *R. glutinis* [1,19], and secondary metabolites produced by *Aspergillus niger* [20]. The proposed mechanisms in biological control of BCAs against BFB include competition for nutrients and space [13,17], production of antibacterial substances [17,18], and induced systemic resistance [16,21].

Strain YL-1 was isolated from the soybean rhizosphere and identified as *P.*
*chlororaphis* [22]. Previous studies showed that this strain has broad-spectrum antibacterial activities against plant pathogens that are economically important in agriculture [23]. Bioinformatics analysis of the genome sequence (accession number: NZ_AWWJ00000000.1) revealed the presence of gene clusters involved in the synthesis of achromobactin, hydrogen cyanide, phenazine, pyoverdine, and pyrrolnitrin, which may play important roles in antimicrobial activities [22]. Pyoverdines are fluorescent yellow-green siderophores produced by fluorescent pseudomonads under iron-limited conditions [24]. Our previous results showed that pyoverdines are essential for the antibacterial activity of *P. chlororaphis* YL-1 under low-iron conditions [25]. Achromobactin is another siderophore produced with pyoverdine in pseudomonads, contributing to iron uptake and growth of its host bacteria [26]. Pyrrolnitrin (PRN) is a pyrrole halometabolite produced by rhizospheric pseudomonads and is known to exhibit biological control against a wide range of phytopathogenic fungi [27]. Hydrogen cyanide (HCN), known as a broad-spectrum antimicrobial compound, has been found to be implicated in the suppression of plant root diseases [28]. Among the putative antibiotics produced by *P.*
*chlororaphis* YL-1, phenazine compounds have been among the most studied [29]. The phenazines, produced by *Pseudomonas* spp. and a few other bacterial genera, include upward of 50 pigmented, heterocyclic nitrogen-containing secondary metabolites, such as phenazine-1-carboxylic acid (PCA), phenazine-1-carboxamide (PCN), 2-hydroxyphenazine-1-carboxylic acid (2-OH-PCA) or 2-hydroxyphenazine (2-OH-PHZ) [29]. Phenazine compounds produced by plant-beneficial *Pseudomonas* spp. display broad-spectrum antibiotic activities toward many fungal, bacterial, and oomycete plant pathogens. Moreover, phenazine production also plays a role in the survival and persistence of plant-beneficial *Pseudomonas* spp. in the rhizosphere and in many physiological processes, including biofilm formation and iron reduction [30], such as *P. synxantha* (formerly *P. fluorescens*) 2-79 [31] and *P. chlororaphis* (formerly *P.*
*aureofaciens*) 30-84 [32]. These results set the stage for improving the performance of phenazine producers used as biological control agents for soil-borne plant pathogens.

This study focused on the main antimicrobial substance by determining the effect on *A. citrulli* in vitro and its mechanism. The results revealed that PCA produced by strain YL-1 is responsible for its antibacterial activity against *A. citrulli*. We also found that PCA enhanced the accumulation of ROS and the activity of CAT in *A. citrulli*.

## 2. Materials and Methods

### 2.1. Microorganisms, Media, and Culture Conditions

Bacterial strains used in the study are listed in Appendix A. *P. chlororaphis* YL-1, the mutants, and three phytopathogenic bacteria (Appendix A) were routinely cultured at 28 °C on Luria-Bertani (LB) medium and preserved in 20% glycerol at −80 °C in long-term storage. *Escherichia coli* S17-1 for bacterial conjugation was routinely cultured at 37 °C on LB medium. To obtain a large amount of PCA, *P. chlororaphis* YL-1 and its mutants were grown in King’s B medium [33]. Standard PCA (STD) sample was provided by Dr. He Lab (Shanghai Jiao Tong University, Shanghai, China). All chemicals were purchased from Sangon (Shanghai, China); unless otherwise stated, the solvents used for extraction were of analytical grade and those for LC-MS of HPLC grade.

### 2.2. Generation of Deletion Mutants

The gene products PhzE and PhzF are conserved in all phenazine-producing bacteria, leading to PCA production in *Pseudomonas* spp. [30]. PrnB is the second enzyme in the pyrrolnitrin biosynthesis pathway [34]. HcnB, an amino acid oxidase, is essential for cyanide production [35]. PvdL, a nonribosomal peptide synthetase (NRPS), is involved in the biosynthesis of pyoverdine [36]. Achromobactin is a siderophore assembled by NRPS-independent siderophore (NIS) synthetases, different from pyoverdine. Both AcsD and AcsA were NIS synthetases [37]. Here, we selected seven above conserved genes, *phzE*, Δ*phzF,* Δ*prnB*, Δ*hcnB*, Δ*pvdL*, Δ*acsA*, and Δ*acsD* to generate in-frame deletion mutants.

All primers used for PCR and cloning are listed in Appendix A. The *B. subtilis sacB* gene, which encodes levansucrase that is toxic to Gram-negative bacteria in the presence of sucrose, was considered as a counter-selection marker in the system [38]. We selected *phzE* as a representative example to generate an in-frame deletion mutant (Appendix A). First, the upstream and downstream homolog fragment of *phzE* in wild-type strain YL-1 was cloned by using two pairs of primers (*phzE*-F1 and *phzE*-R1, *phzE*-F2 and *phzE*-R2). Then, a *sacB*-containing suicide vector, pEX18Gm [39] was used to create a recombinant plasmid, pEX18- *phzE*. The recombinant plasmid was first transformed into *E. coli* S17-1 [40]; then, it was transformed into wild-type strain YL-1 via an optimized bacterial conjugal approach. Via a double-crossover homologous recombinant approach, an in-frame deletion mutant of *phzE*, named as Δ*phzE*, was generated and validated [41]. The mutants Δ*phzF,* Δ*prnB*, Δ*hcnB*, Δ*acsA*, and Δ*acsD* were generated as described above.

Gene replacements and insertions were verified by PCR amplifications using specific primers and DNA sequencing. For example, using genomic DNA samples of the mutant and wild-type strain as PCR templates, respectively, specific primers (Ex. *phzE*-F1and *phzE-*R2, Appendix A) were used to amplify the target region. The resulting PCR product with different sizes was observed using agarose gel electrophoresis. Then, the two PCR products were sequenced, and sequence analysis was conducted to verify the authenticity of mutants.

### 2.3. Growth, Hydrogen Cyanide and Pyoverdine Measurements

*P. chlororaphis* YL-1 and its seven mutants (Appendix A) were individually incubated in a 250 mL flask containing 100 mL liquid LB medium or low-iron succinic acid medium (SM) [42] at 28 °C with shaking (200 rpm for 24 h). All liquid cultures were observed under visible light and ultraviolet light simultaneously. Colony-forming units per mL (CFU/mL) were determined by the standard serial dilution. The experiment was repeated three times independently.

Production of hydrogen cyanide was observed according to the method of Lorck [43]. Freshly grown cells were spread on King’s B medium containing glycine (4.5 g/L). A sterilized filter paper saturated with 1% solution of picric acid and 2% sodium carbonate was placed in the upper lid of a petri dish. The petri dish was then sealed with parafilm and incubated at 30 °C for 4 days. A change in color of the filter paper from yellow to reddish brown as an index of cyanogenic activity was recorded.

### 2.4. Antibacterial Activity of Strain YL-1 and Mutants

Strain YL-1 and three phytopathogenic bacteria were incubated in LB media at 28 °C with shaking (200 rpm for 24 h). Thereafter, the cultures were resuspended in LB broth to the desired cell density of 10^8^ CFU/mL. Cells of three phytopathogenic bacteria, *A. citrulli* strain XJX12, *Xanthomonas oryzae* pv. *oryzae* (*Xoo*) strain PXO99, and *X. oryzae* pv. *oryzicola* (*Xooc*) strain RS11, were evenly sprayed onto the surface of LB plates for 1 s using sterile laryngeal spray (Taizhou Medical Instrument Co. Ltd., Taizhou, China). Then, 5 μL cells of strain YL-1 (10^8^ CFU/mL) were added on the LB plates, 1 cm from the edge of the plate in a cross shape. Distilled LB broth was used as a control. Each treatment was replicated three times and each experiment was repeated thrice independently. The inhibitory zone diameters of strain YL-1 were measured when phytopathogenic bacteria covered the control plates entirely.

The differences in antibacterial activities between YL-1 and its seven mutants were evaluated as described above [25]. After phytopathogenic bacteria (*A. citrulli* or *Xoo*) were evenly sprayed onto the surface of LB plates, 5 μL cells of YL-1 or mutants (10^8^ CFU/mL) were injected into the central hole (5 mm) of LB plates and incubated at 28 °C for 48 h. Distilled LB broth was used as a control. The inhibitory zone diameters of strain YL-1 and mutants were measured when *A. citrulli* or *Xoo* covered the control plates entirely.

### 2.5. Extraction and Determination of PCA

To extract the maximum PCA, strain YL-1 and mutants Δ*phzE* and Δ*phzF* were cultured in 100 mL liquid King’s B medium at 28 °C for 48 h, then culture supernatants were mixed with 15 mL of 6 M HCl and extracted with 100 mL chloroform as previously described [44]. The soluble organic fraction was concentrated by a rotary evaporator at a temperature below 35 °C, 70 mg of crude compound was obtained and dissolved in 7 mL acetone, and then the concentration was 10 mg/mL. A three-microliter aliquot of extracted PCA from the fermentation broth of strain YL-1 and STD sample was then taken for HPLC analysis (Agilent Technologies 1260 Infinity) under the following conditions: XDB-C18 reversed-phase column (4.6 mm × 150 mm, 5 μm, Agilent) eluted with acetonitrile (ACN) −5 mM ammonium acetate (60:40, *v*/*v*) at a flow rate of 0.7 mL/min.

Thereafter, the fraction from retention time 1.95 to 2.10 min was first collected and concentrated by a rotary evaporator, then dissolved in acetone and determined with ultra-performance liquid chromatography coupled with mass spectrometry (Agilent UPLC 1290-MS 6230, LC/MS) under the following conditions: Zorbax XDB C18 reverse-phase (4.6 mm × 150 mm, 5 μm, Agilent) eluted with gradient H_2_O with 0.5% acetic acid (A) and ACN with 0.5% acetic acid (B). Mobile phase flow rate was set at 0.4 mL/min using a linear gradient of A and B starting at 0% B, increasing to 50% B in 5 min, increasing to 100% B in 8 min, maintained at 100% for 10 min, and returned to 0% B in 2 min. Total run time was 28 min including 3 min equilibration time at the end of the gradient. The MS analysis was performed under positive mode with a scanning range of m/z = 100–1700.

### 2.6. Inhibition of PCA against A. citrulli

Inhibition of PCA against *A. citrulli* was determined as previously described [45]. Strain XJX12 was cultured to late-log phase (OD_600_ of approximately 1.0) in LB medium at 28 °C with shaking (200 rpm for 24 h). The bacterial suspension was diluted 1, 10, and 50 times using sterile liquid LB broth. A 2 μL volume of diluted bacterial suspension was spotted onto LB plates containing an acetone solution without or with PCA to achieve final PCA concentrations of 0, 2, 4, 8, 16, or 32 μg/mL and a final acetone concentration of 0.4% (*v*/*v*). The growth of *A. citrulli* cells was examined after 3 d at 28 °C. Each treatment was represented by three replicates, and the experiment was repeated three times.

Meanwhile, a 1 mL volume of bacterial suspension (OD_600_ of approximately 1.0) was then added to 100 mL of fresh liquid LB medium containing PCA at 0, 2, 4, 8, 16, or 32 μg/mL (final acetone concentration 0.4% (*v*/*v*)). Strain XJX12 was grown at 28 °C with shaking (200 rpm for 12 h) before turbidity values were determined (2100-N Scattering Turbidimeter, Hach). The average turbidity values were used to calculate the PCA concentration that resulted in 50% inhibition of bacterial cell growth (EC_50_). The EC_50_ values were calculated with the Data Processing System computer program (Hangzhou Reifeng Information Technology Ltd.). Each treatment was replicated three times and each experiment was repeated thrice independently.

### 2.7. Transcript Levels of Transcriptional Regulator Genes as Affected by PCA

Strain XJX12 was grown in liquid LB medium, containing acetone solution without or with PCA. The concentration of PCA was 16 or 32 μg/mL. The final acetone concentration was 0.4% (*v*/*v*). RNA isolation and quantitative reverse-transcription polymerase chain reaction (qRT-PCR) were performed as previously described [46]. Strain XJX12 was grown to OD_600_ values of 0.1 (with culture time 6 h), 0.2 (with culture time 8 h), 0.35 (with culture time 10 h), 0.5 (with culture time 12 h), 0.6 (with culture time 14 h), and 1.0 (with culture time 24 h), respectively, and a 2 mL volume of bacterial suspension was harvested by centrifugation (8000× *g* for 2 min at 4 °C). Total RNA was extracted using a total RNA Miniprep Purification Kit (Sangon Biotech, Shanghai). Total RNA was reverse-transcribed using PrimeScript^TM^ RT Master Mix (TaKaRa, Japan). qRT-PCR was performed on a Roche Lightcycler 2.0 System (Switzerland) using Power TB Green PCR Master Mix (TaKaRa, Japan). The conserved *XJ-RT* gene of *A. citrulli* (accession number: CP000512.1), encoding amino acid adenylation domain protein, was used as an internal control in the qRT-PCR assay [47]. Each 20 μL RT-PCR reaction volume contained 1 ng cDNA, 5× TB Green Mix, and 4 μM of forward and reverse primers (Appendix A). The *C_T_* values of *oxyR* or *soxR* used to determine transcript levels were normalized to those of the *XJ-RT* gene. Three replicates of RT-PCR analysis for each reaction were performed and repeated independently.

In addition, gene expression of *katB*, *katE*, *katG* (encoding CAT), *trxA* (encoding thioredoxin (Trx)), *ahpC*, *ahpF* (encoding alkyl hydroperoxide reductase subunits CF (AhpCF)), *sodA, sodB*, and *sodC* (encoding SOD), which are associated with oxidative stresses [46], was assessed using the same cDNA template as described above.

### 2.8. ROS Accumulation, Activities of Superoxide Dismutase (SOD) and CAT in A. citrulli Induced by PCA

ROS accumulation was measured with a ROS assay kit (Beyotime, China) as previously described [46]. In brief, bacterial cultures of OD_600_ values 1.0 were collected by centrifugation (6000× g, 10 min) and resuspended in 2′,7′-dichlorodihydrofluorescein diacetate (DCFH-DA) solution, then incubated at 28 °C for 20 min. After being washed three times, the cells were resuspended in XOM medium (a compound mixture, provided by the assay kit) and added to microplates. The wells were treated with an acetone solution without or with PCA to achieve final PCA concentrations of 0, 16, or 32 μg/mL and a final acetone concentration of 0.4% (*v*/*v*). The microplates were kept at 28 °C, and an Infinite^®^ M1000 Pro fluorescence microplate reader (Tecan, Switzerland) was used to record the fluorescence for 4 h with an excitation wavelength of 488 nm and an emission wavelength of 525 nm. Each treatment was represented by three replicates, and the experiment was conducted three times.

SOD and CAT activities were assessed as previously described [46]. Strain XJX12 was grown in liquid LB medium, containing acetone solution without or with PCA. The concentration of PCA was 16 or 32 μg/mL. The final acetone concentration was 0.4% (*v*/*v*). At different time points, bacterial cells of strain XJX12 were treated with 200 μL of Cell Lysis Liquid (Beyotime, China). The protein concentration in the supernatant was determined with a BCA Protein Determination Kit (Beyotime, China), and total CAT and SOD activities were determined with the CAT Test Kit and SOD Test Kit (Beyotime, China). The absorption values were recorded with visible spectrophotometer Epoch2 (BioTek). Each treatment was represented by three replicates, and the experiment was conducted three times.

### 2.9. Sensitivity of A. citrulli as Affected by Exogenous H_2_O_2_

Strain XJX12 was cultured in LB medium at 28 °C with shaking (200 rpm for 24 h, OD_600_ of approximately 1.0). The bacterial suspension was diluted 1, 10, and 50 times as described above. A 2 μL volume of diluted bacterial suspension was spotted onto LB plates containing 0, 0.25, 0.5, 0.75, 1.0, or 1.25 mM H_2_O_2_. The growth of *A. citrulli* cells was examined after 3 d at 28 °C. Each treatment was represented by three replicates, and the experiment was repeated three times.

### 2.10. Inhibition of PCA against A. citrulli as Affected by Exogenous CAT

A 1 mL volume of bacterial suspension (OD_600_ of approximately 1.0) was added to 100 mL of fresh LB medium, which was then supplemented with acetone or acetone containing PCA to achieve a final PCA concentration of 16 or 32 μg/mL in strain XJX12 culture and a final acetone concentration 0.4% (*v*/*v*). After that, 0, 25, 50, 75, 100, 150, or 200 U of CAT were added to the medium. The cultures were kept at 28 °C with shaking at 200 rpm, and turbidity values were determined after 12 h. The effect of CAT on PCA inhibition against *A. citrulli* was expressed as a percentage of growth inhibition. Each treatment was represented by three replicates, and the experiment was repeated three times.

### 2.11. Statistical Analysis

Data were processed by analysis of variance (ANOVA) using the SAS GLM procedure (SAS Institute, Inc., Cary, NC, USA). *p* < 0.05 was considered statistically significant. Significant means were further compared using Fisher’s protected least significant difference (PLSD).

## 3. Results

### 3.1. Growth, Hydrogen Cyanide and Pyoverdine Production of P. chlororaphis YL-1 and Mutants

As expected, no difference was observed in the fermentation broth color or the growth capacity between *P. chlororaphis* YL-1 and its seven mutants in LB medium. Likewise, strain YL-1 and its mutants in LB medium did not exhibit any fluorescence under UV light (data not shown). Fermentation broths of strain YL-1, mutants Δ*phzE,* Δ*phzF,* Δ*prnB,* Δ*hcnB,* Δ*acsA,* and Δ*acsD* in SM medium were yellow green in visible light and exhibited fluorescence under UV light, which were the special characteristics of pyoverdine produced by *Pseudomonas* spp. (Appendix A); however, fermentation broth of the mutant Δ*pvdL* was white and did not exhibit fluorescence under UV light (Appendix A). The growth capacity of all the mutants in SM medium was comparable to that of the wild-type strain.

A remarkable change in color from yellow to reddish-brown of strain YL-1 and mutants Δ*phzE,* Δ*phzF,* Δ*prnB,* Δ*pvdL,* Δ*acsA,* and Δ*acsD* compared with the mutant Δ*hcnB* indicates the production of hydrogen cyanide (Appendix A). These results showed that the *sacB*-mediated genetic manipulation system used for *P. chlororaphis* YL-1 was stable and efficient.

### 3.2. Antibacterial Activity of P. chlororaphis YL-1 and Mutants

Antibacterial activities of strain YL-1 and mutants precultured in liquid LB medium were determined on LB plates. Strain YL-1 produced clear inhibitory zones on anti-*A. citrulli* and anti-*Xoo* activities (Appendix A). The inhibitory zone against *A. citrulli* was 32.5 ± 1.8 mm while that against *Xoo* was 42.3 ± 1.7 mm in diameter. There was a slight inhibitory effect on *Xooc* by strain YL-1 on LB plates with inhibitory zone 9.7 ± 2.2 mm.

Strain YL-1 and five mutants, Δ*prnB,* Δ*hcnB,* Δ*pvdL,* Δ*acsA,* and Δ*acsD*, produced clear inhibitory zones with no observable difference in anti-*A. citrulli* or anti-*Xoo* activities. Inhibitory zones against *A. citrulli* ranged from 55.8 ± 2.3 mm to 66.1 ± 1.2 mm (Figure 1), and those against *Xoo* ranged from 70.2 ± 1.5 mm to 75.3 ± 2.1 mm in diameter (Appendix A). However, there was little inhibitory effect on *A. citrulli* or *Xoo* treated by mutants Δ*phzE* and Δ*phzF* (Figure 1, Appendix A). These results suggest that phenazine is the key metabolite in antibacterial activity against *A. citrulli*.

### 3.3. Isolation and Purification of Phenazine

In this study, it was found that one comparable peak was produced by STD sample (blue line) and the wild-type strain YL-1 (pink line) under the absorbance wavelength of 254 nm, with the retention time 2.0 min (Figure 2a). No obvious peak was detected from the samples of the mutants Δ*phzE* (green line) and Δ*phzF* (black line) (Figure 2b). A comparable peak was also detected at a higher level in STD sample, compared to that extracted from strain YL-1 (Figure 2a). The fraction collected from 1.95 to 2.10 min was further purified and analyzed by LC-MS. As shown in Figure 3, the characteristic ions of the mass spectrum in strain YL-1 are also consistent with those in STD sample, with m/z of 225.07, 226.07, and 227.07 (three isotopic peaks). The main antibacterial compound produced by strain YL-1 in LB medium was identified as PCA.

### 3.4. The Inhibition of PCA against A. citrulli

As shown in Figure 4, colony growth of *A. citrulli* was inhibited when PCA concentration in LB plates reached 8 μg/mL. *A. citrulli* can grow well on LB plates containing 4 μg/mL PCA even if diluted 50 times. However, it can hardly grow on LB plates containing 32 μg/mL PCA.

The inhibition of PCA against *A. citrulli* was more sensitive in liquid culture than that on LB plates (Figure 5). The inhibition rates were 10.96%, 19.77%, 45.18%, or 66.11% when PCA concentration in liquid LB medium reached 4, 8, 16, or 32 μg/mL, respectively. The linear regression equation was y = 2.7545 + 1.7315x and the EC_50_ value of PCA was 19.81 μg/mL.

### 3.5. Expression of oxyR, soxR, and the Genes Involved in Antioxidant Systems as Affected by PCA

To determine whether transcriptional regulators SoxR and OxyR were activated by PCA, and to determine whether the genes encoding antioxidant enzymes were also induced by PCA in *A. citrulli*, expression of the genes of interest was performed using qRT-PCR. When the concentrations of PCA were 16 or 32 μg/mL, the expression levels of *oxyR* were 1.33 or 2.13 times higher at 6 h (Figure 6a), and 1.84 or 3.31 times higher at 8 h (Figure 6b) than those of the acetone control, respectively. Then, the expression level of *oxyR* decreased to control level at 10 h (Figure 6c). The expression levels of *katB*, *ahpC*, and *ahpF* regulated by OxyR began to increase at 6 h after PCA treatment. At 8 h, treated by 16 or 32 μg/mL PCA, the expression levels of *katB*, *ahpC*, and *ahpF* were 8.00 or 20.46 times higher, 1.49 or 3.26 times higher, and 1.49 or 2.14 times higher than those of the acetone control, respectively (Figure 6b). The expression level of *katG* was 1.65 or 2.67 times higher than that of the control at 6 h treated by 16 or 32 μg/mL PCA, then it began to decrease sharply. The expression of other genes (*katE* and *trxA*) had no significant change from 6–10 h after being treated by PCA. The expression level of *soxR*, encoding another transcription regulator SoxR, increased at 6 h after being treated by 16 or 32 μg/mL PCA, being 1.46 or 1.64 times higher than that of the control (Figure 6a), then decreasing to control level at 8 h. However, the expression of *sodA*, *sodB*, and *sodC* remained unchanged from 6–10 h after being treated by PCA. These results indicate that *katB* in *A. citrulli* plays an important role in the process of protecting the bacteria against PCA-induced oxidative stress.

### 3.6. ROS Accumulation, CAT and SOD Activities Induced by PCA

ROS levels in *A. citrulli* increased significantly when treated with PCA or Rosup reagent (the positive control) (Figure 7a). ROS accumulation was greater with 16 or 32 μg/mL PCA than with Rosup reagent. As expected, ROS accumulation was not detected in the negative control or acetone treatment.

As shown in Figure 7b, CAT activity in *A. citrulli* was found to increase from 6 h to 12 h after being treated by acetone, 16 or 32 μg/mL PCA, with the maximum value 12.53, 13.09, or 15.02 U at 8 h, respectively. Then, CAT activity was decreased to normal level at 14 h. No significant difference was found for SOD activity treated by acetone, 16 or 32 μg/mL PCA (data not shown). These data suggest the *A citrulli* cells enhanced CAT activity to respond to the PCA treatment.

It should be noted that acetone (the solvent of PCA) reaching a certain concentration would have an inhibitory effect on *A. citrulli* (data not shown); therefore, the concentration of acetone in the research study was set as 0.4% (*v*/*v*). *A. citrulli* was not inhibited and can grow well on LB plates (Figure 4) and in LB liquid medium (Figure 5) containing 0.4% acetone (*v*/*v*). Although 0.4% acetone has a slight influence on CAT activity in *A. citrulli* (Figure 7), significant difference was found as compared with 32 μg/mL PCA from 6 h to 10 h.

### 3.7. Sensitivity of A. citrulli as Affected by Exogenous H_2_O_2_

Strain XJX12 was sensitive to H_2_O_2_ and did not grow on LB plates containing 1.5 mM H_2_O_2_. It grew well at 0.25 mM even if diluted 50 times. The growth of strain XJX12 was inhibited on LB plates containing 0.5 mM H_2_O_2_ when diluted 50 times, and those containing 1.0 mM H_2_O_2_ when diluted 10 times (Appendix A). These data demonstrate that bacterial cells are sensitive to the accumulation of active oxygen.

### 3.8. Inhibition of PCA against A. citrulli as Affected by Exogenous CAT

The inhibition effect of PCA against *A. citrulli* was lowered by adding exogenous CAT. As shown in Figure 8, when treated with the same concentration of PCA (32 μg/mL), decreases in inhibition effect correlated to increases in the exogenous CAT content of the medium, with inhibition rates of 66.10 ± 7.93, 58.64 ± 7.03, 50.24 ± 6.02, 44.40 ± 5.32, 34.96 ± 4.19, 20.07 ± 2.41, and 9.42 ± 1.13 when treated with 0, 25, 50, 75, 100, 150, and 200 U CAT in liquid LB medium, respectively. As shown in Figure 8, the inhibition effect upon treatment with 16 μg/mL of PCA was substantially weaker than that upon treatment with 32 μg/mL at any concentration of exogenous CAT.

## 4. Discussion

Genes responsible for phenazine biosynthesis are organized in a seven-gene operon *phzABCDEFG*, which is conserved in all phenazine-producing *Pseudomonas* spp. [29]. The first enzyme involved in phenazine biosynthesis is a type II 3-deoxy-D-arabinoheptulosonate-7-phosphate (DAHP) synthase, PhzC. Five enzymes—PhzE, PhzD, PhzF, PhzB, and PhzG—are conserved in all phenazine-producing bacteria, leading to PCA production in *Pseudomonas* spp., but the same enzymes can also produce phenazine-1,6-dicarboyxlic acid (PDC) in other phenazine-producing bacteria [30]. The gene *phzA*, present in *Pseudomonas* spp., has a high homology to *phzB* (≈70% identity) and is important in PCA production [30]. Some phenazine-producing *Pseudomonas* spp. harbor accessory phenazine biosynthetic genes, such as *phzH* and *phzO*, which are located immediately after *phzG*. These genes encode phenazine-modifying enzymes that are responsible for the production of strain-specific phenazine derivatives [30]. Bioinformatics analysis of the genome sequence (accession number: NZ_AWWJ00000000.1) revealed the presence of *phzABCDEFGO* gene clusters in *P. chlororaphis* YL-1. We selected two conserved genes, *phzE* and *phzF*, to generate in-frame deletion mutants; the mutants Δ*phzE* and Δ*phzF* did not produce PCA and lost their antibacterial activity compared with wild-type strain YL-1. Surprisingly, complementation of the mutants, using the functional *phzE* and *phzF* gene fragments harbored in the expression vector pUCP26 [48], failed, although considerable efforts were made. In fact, mutations of multiple *phz* genes could not be complemented in numerous studies [49,50,51]. The *Pseudomonas* expression vector pUCP26 has been successfully used for genetic complementation of mutations in *Pseudomonas* [25,52]. It remains to be investigated if there are any effects of copy number on gene expression in this case. As expected, gene-specific deletions in the mutant Δ*phzE* and Δ*phzF,* which were generated by the *sac B*-based counter-selection marker mutagenesis approach [38], were confirmed by sequencing analysis (Appendix A). Nevertheless, the results suggest that PCA is the main antibacterial phenazine compound produced by stain YL-1.

Secondary metabolites produced by a microorganism and its antimicrobial activities can vary depending on the culture media used for growth. Previous results showed that the main antibacterial secondary metabolite produced by strain YL-1 is pyoverdine in low-iron condition. Compared with wild-type strain YL-1, pyoverdine-deficient mutants cannot produce pyoverdine and their antibacterial activity reduced significantly when grown on low-iron medium [25]. However, the antibacterial activity of PVD-deficient mutants did not change in iron-sufficient conditions (such as LB medium) compared with that of wild-type strain YL-1 [25]. These results suggest that strain YL-1 and PVD-deficient mutants secrete different antibacterial compounds when cultured on LB plates as compared with low-iron conditions. In this study, we evaluated the antimicrobial activity of strain YL-1 on LB plates as compared with its phenazine-deficient mutants Δ*phzE* and Δ*phzF*, pyrrolnitrin-deficient mutant Δ*prnB*, hydrogen-cyanide-deficient mutant Δ*hcnB*, pyoverdine-deficient mutant Δ*pvdL*, and achromobactin-deficient mutants Δ*acsA* and Δ*acsD*. The results showed that only phenazine-deficient mutants Δ*phzE* and Δ*phzF* almost lost the inhibitory effects against *A. citrulli* and *Xoo* on LB plates compared with strain YL-1 (Figure 1 and Appendix A), and that the main antibacterial compound produced by strain YL-1 on LB plates was PCA, based on LC-MS/MS analysis (Figure 2 and Figure 3). Later, it was also verified that purified PCA produced by strain YL-1 can inhibit the growth of *A. citrulli* in LB plates and in liquid culture (Figure 4 and Figure 5). These findings of the research further demonstrate that *P. chlororaphis* YL-1 can produce multiple antimicrobial compounds to target various microbial organisms under different conditions.

According to the reports of comparative genomic and core genome analyses of the various *Pseudomonas* groups, phenazine biosynthetic gene *phzF* is present in 86% (37/43) of all tested *P. chlororaphis* strains while as a core gene it exists in the analyzed *P. aeruginosa* strains (189/189). The *pvdL* gene was identified as a core gene in all tested *P. chlororaphis* strains (43/43) while it exists in 90% (171/189) of the analyzed *P. aeruginosa* strains and 81% (212/262) of all the other *Pseudomonas* strains [53]. These results indicate that both phenazine and pyoverdine biosynthesis genes are distributed widely across the *Pseudomonas* genus. In addition, it was reported that production of antifungal compounds phenazine and pyrrolnitrin in *P. chlororaphis* O6 is differentially regulated by glucose [54]. More research is needed to investigate any possible variation of PCA production under a different medium. Moreover, it is worth noting that PCA is categorized as an irritant in PubChem toxicology databases and it will be harmful to eyes, skin, and the respiratory tract after inappropriate operation.

PCA is a natural product that is isolated from *Pseudomonas* spp. and is used to control many rice diseases in China, such as rice sheath blight, rice blast, rice bacterial leaf blight, and rice bacterial leaf streak [55]. PCA also exhibits extensive antimicrobial activities against phytopathogens in vitro, such as *Xoo*, *Xooc*, *Rhizoctonia solani*, *Alternaria solani*, *Fusarium oxysporum*, *Fusarium graminearum*, and *Pyricularia oryzac*, with EC_50_ values of 0.18, 13.11, 7.44, 18.26, 41.78, 39.51, and 8.36 μg/mL, respectively [55,56]. To our knowledge, there was only one report that PCA, produced by a marine bacterium *P. aeruginosa* PA31x, can inhibit the growth of *A. citrulli* in vitro*,* with the inhibitory efficiency 70% when treated by 10 mg/mL of PCA [57]. However, the mechanism of PCA against *A. citrulli* remains unclear. In this study, *A. citrulli* can hardly grow on LB plates and the inhibitory efficiency in the liquid coculture test was 66.11% when treated by 32 μg/mL of PCA. The EC_50_ value of PCA against *A. citrulli* was 19.81 μg/mL. Moreover, we also found that PCA enhanced the accumulation of ROS and increased the activity of CAT in *A. citrulli*. The inhibition of PCA against *A. citrulli* was reduced by adding exogenous CAT (Figure 8), indicating that PCA can interfere with electron transport chains, generating ROS, damaging the target bacteria, and ultimately leading to cell death in bacterial cells [58]. *P. chlororaphis* YL-1 can inhibit the growth of both Gram-positive and Gram-negative bacteria, such as *Burkholderia glumae*, *Clavibacter michiganesis*, and *Erwinia amylovora* [22]. To discern whether PCA produced by strain YL-1 is also the main antibacterial secondary metabolite against Gram-positive bacteria and whether its mechanism is the same as those in Gram-negative bacteria, further investigation is needed.

Two key transcriptional regulators, SoxR and OxyR, have been well known to respond to oxidative stress and regulate antioxidant-related enzymes in most bacteria. SoxR is activated in response to O_2_^-^ and, in turn, activates SOD. In contrast, OxyR is activated in response to H_2_O_2_ and, in turn, activates CAT, Trx, and AhpCF [46]. In this study, we found that expression levels of *katB*, *katG*, and *ahpCF* regulated by transcription regulator OxyR in *A. citrulli* increased 6 h after being affected by PCA. Among them, the most significant one is *katB,* with expression levels 8.00 or 20.46 times higher than those of the acetone control 8 h after being treated by 16 or 32 μg/mL PCA, respectively (Figure 6b). Similarly, expression levels of *catB* in *Xoo* were 18.24 times higher than those of the acetone control when treated by 4 μg/mL PCA while they were 6.63 times higher in *Xooc* when treated by 32 μg/mL PCA [46]. However, the expression of *katE* and *trxA* in both *Xoo* and *Xooc* increased when affected by PCA [46], while not being changed in *A. citrulli*. The expression of *sodA*, *sodB*, and *sodC* in *A. citrulli* regulated by SoxR remained unchanged when affected by PCA. PCA only induced the expression of *sodM1* in *Xoo* while PCA did not induce the expression of *soxR* or the genes encoding SOD in *Xooc* [46]. These results show that different bacteria have various CATs that are encoded and activated by different genes although the accumulation of ROS in *A. citrulli* affected by PCA is similar to that in *Xoo* [55]. Currently there is no fully clarified association network of related genes in response to oxidative stress in the above described bacteria, but there is evidence that expression intensity of these different genes determines the difference in antioxidant capacity of bacteria. For example, it was reported that *Xooc* has a stronger antioxidant system than *Xoo* and accumulates lower ROS in response to redox compounds by deleting or exchanging the antioxidant-related genes, although *Xoo* and *Xooc* are very closely related [55]. Therefore, our results provide a scientific basis for the study of antioxidant capacity, virulence, and control of *A. citrulli* in the future.

Moreover, the different genes encoding antioxidant enzymes as described above possess different functions at different growth phases. In a rhizobium, *Azorhizobium caulinodans*, KatG and OxyR are not only critical for antioxidant defense in vitro, but are also important for nodule formation and nitrogen fixation during interaction with plant hosts [59]. *P. aeruginosa* possesses an extensive armament of genes involved in oxidative stress defense, including *katB*-*ankB*, *ahpB*, and *ahpC*-*ahpF*. Expression of these genes was not observed in Δ*oxyR* mutant, indicating that OxyR was essential for this response. Moreover, increased *katB* expression and higher CAT levels were detected in Δ*ahpCF*, suggesting a compensatory function for KatB in the absence of AhpCF [60]. Cyclic dimeric guanosine monophosphate (c-di-GMP), as a dinucleotide second messenger, increased CAT activity and expression of the CAT encoding gene *katB* in *Vibrio cholerae* [61]. The monofunctional CAT, KatE, of *X. axonopodis* pv. *citri* is not only important for its colonization and survival in citrus plants, but also is required for full virulence in citrus plants [62]. In a variety of bacteria, including mycobacteria, KatG is a unique dual function enzyme acting as CAT and peroxidase, which are important for protection against damage from ROS [63]. SOD isozymes in *Klebsiella pneumonia* have unique roles beyond oxidative stress resistance, such as biofilm formation, cell morphology, metabolism, and in vivo colonization and persistence and there is a regulatory interplay among SODs [64]. The *FurA* gene, which codes for a ferric uptake regulator, is required for *A. citrulli* virulence on watermelon; *Ac*Δ*furA* displayed increased sensitivity to H_2_O_2_ and significant downregulation of *sodB* [47]. As far as we know, the transcriptional regulator and related genes involved in oxidative stress defense were seldom reported in *A. citrulli*. Therefore, it remains to be investigated to explore more functions of antioxidant-related enzymes in *A. citrulli*.

In conclusion, the findings of this research demonstrate that strain YL-1 could produce PCA, which has strong inhibitory effects on Gram-negative bacteria as demonstrated by using bacterial cells and purified compounds, when grown on LB medium. PCA can enhance the accumulation of reactive oxygen species (ROS) and the activity of catalase (CAT) in *A. citrulli*, and the inhibition effect of PCA against *A. citrulli* was lowered by adding exogenous CAT. The results suggest that strain YL-1 is a promising biocontrol bacterium and PCA could inhibit the growth of *A. citrulli* in vitro. However, biocontrol bacteria with in vitro antagonistic activity may not effectively control plant diseases in the field. Additional studies regarding the characteristics of PCA, such as stability, the duration of the efficacy, and product formulation, are needed to confirm the biocontrol effect of PCA under field conditions.

## Figures and Tables

**Figure 1 microorganisms-09-02012-f001:**
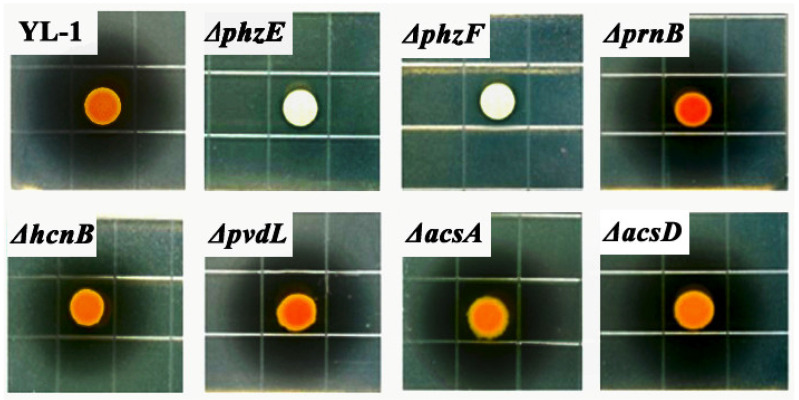
Antibacterial activity of *P. chlororaphis* YL-1 and its seven mutants on LB plates. All strains precultured in liquid LB medium at 28 °C and 200 rpm for 24 h were resuspended in distilled liquid LB medium to the desired cell density of 10^8^ CFU/mL. After spraying the suspension of *A. citrulli* (10^8^ CFU/mL) onto the surface of LB plates for 1 s, 5 μL cells of strain YL-1 or mutants were injected into the central hole (5 mm) of LB plates and incubated at 28 °C for 48 h. Distilled LB medium was used as a control. The inhibitory zone diameters of strain YL-1 and mutants were measured when *A. citrulli* covered the control plates entirely. Each treatment was replicated three times, and each experiment was repeated thrice independently.

**Figure 2 microorganisms-09-02012-f002:**
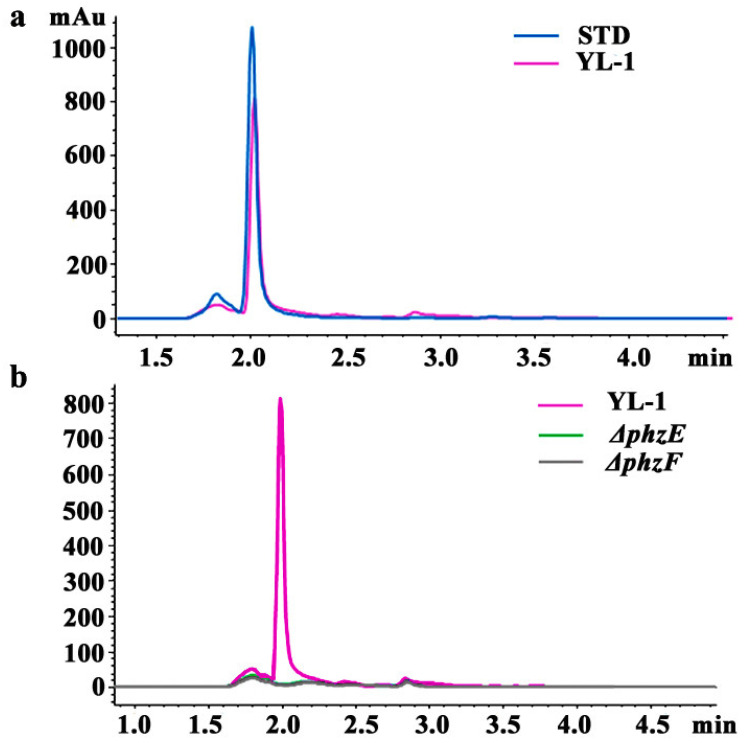
Reversed phase (RP)-HPLC chromatograms. An overlay of the chromatograms at 254 nm of STD sample (blue) and the final extraction step of the wild-type YL-1 (pink) (**a**), mutants Δ*phzE* (green), and Δ*phzF* (black) (**b**), using an XDB-C18 reversed-phase column (4.6 mm × 150 mm, 5 μm, Agilent), is shown. Here, STD sample (blue) was used as a positive control. The samples were eluted with acetonitrile (ACN)-5 mM ammonium acetate (60:40, *v*/*v*) at a flow rate of 0.7 mL/min. The experiment was repeated three times.

**Figure 3 microorganisms-09-02012-f003:**
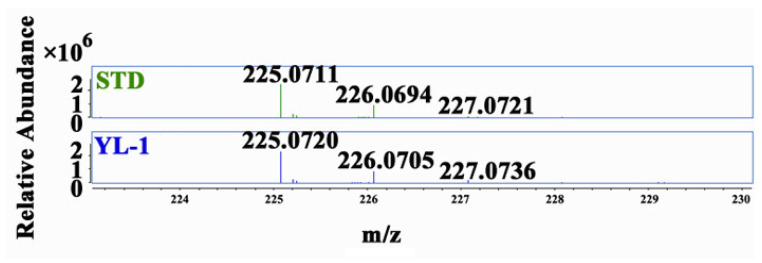
MS spectrum at 2.0 min. The flow rate was maintained at 0.4 mL/min, and the mobile phase comprised H_2_O with 0.5% acetic acid (A) and ACN with 0.5% acetic acid (B) as follows: 0 min, 0% B; 0–5 min, 50% B; 5–13 min, 100% B; 13–23 min, 100% B, 23–25 min, 0% B, and 25–28 min, 0% B. The detecting wavelength was 254 nm and the sample volume injected was 10 µL. The ESI source parameters were set as follows: positive ion mode, 3 kV spray voltage, 350 °C capillary temperature, nitrogen served as both the sheath gas (35 units) and auxiliary gas (10 units). The proposed three characteristic fragment ions of PCA produced by *P. chlororaphis* YL-1 (blue) were consistent with STD sample (blue). The experiment was repeated three times.

**Figure 4 microorganisms-09-02012-f004:**
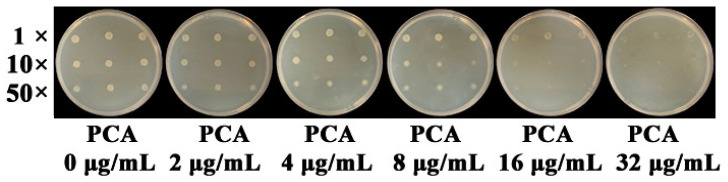
Inhibition of PCA against *A. citrulli* on LB plates. The one-, ten-, and fifty-fold dilutions (1×, 10×, and 50×) of the bacterial suspension were spotted onto LB plates, which contained different concentrations of PCA at 0, 2, 4, 8, 16, and 32 μg/mL. Inoculated plates were incubated at 28 °C for 72 h. Each treatment was replicated three times, and each experiment was repeated thrice independently.

**Figure 5 microorganisms-09-02012-f005:**
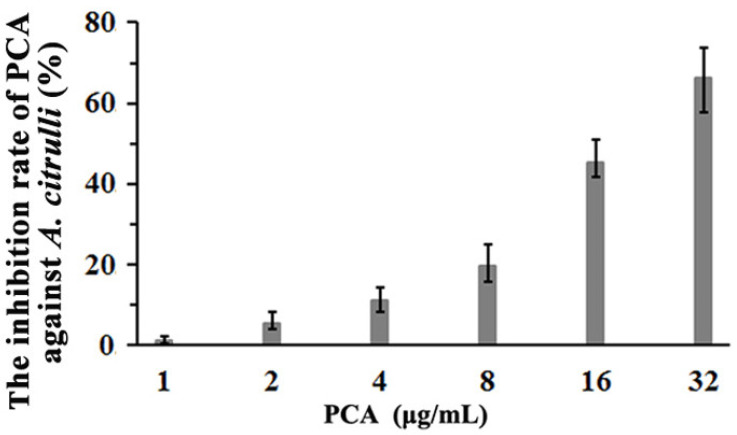
Inhibition effect of PCA at different concentrations on *A. citrulli* when cultured in liquid LB medium (200 rpm; 28 °C for 12 h). The experiment was repeated three times.

**Figure 6 microorganisms-09-02012-f006:**
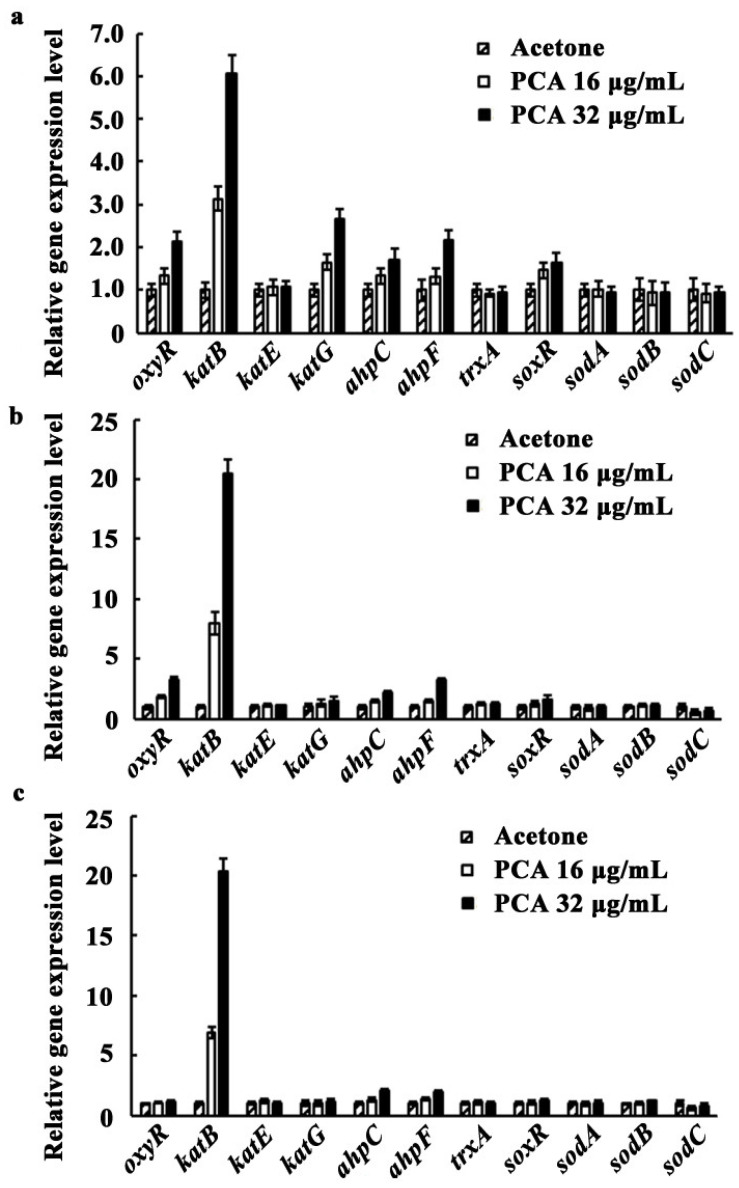
Expression of *oxyR*, *soxR*, and the genes involved in antioxidant systems in *A. citrulli* as affected by PCA. Strain XJX12 was grown in liquid LB medium containing acetone solution without or with PCA to OD_600_ values of 0.1 (with culture time 6 h) (**a**), 0.2 (with culture time 8 h) (**b**), and 0.35 (with culture time 10 h) (**c**). The amount of RNA in acetone was used as the control and was set at 1.0. The *XJ-RT gene* (GenBank number: CP000512.1) was used as an internal control in the qRT-PCR assay. The experiment was repeated three times.

**Figure 7 microorganisms-09-02012-f007:**
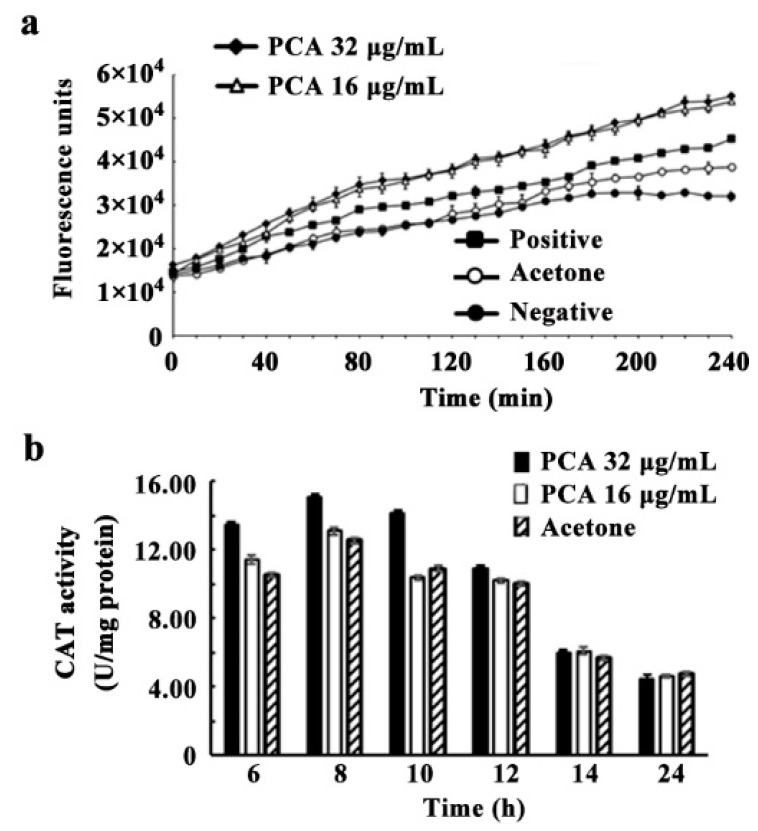
ROS accumulation (**a**) and CAT (**b**) activity of *A. citrulli* strain XJX12 induced by PCA. The treatments included PCA in acetone and acetone alone. ROS accumulation was measured based on fluorescence in microplates, the wells of which contained a bacterial suspension and no additional agent (the negative control) or one of the following: PCA at 0, 16, or 32 μg/mL, with 0.4% (*v*/*v*) acetone (as the solvent) in each case; acetone alone (0.4%, *v*/*v*); or Rosup reagent (a positive control provided with the assay kit). The experiment was repeated three times.

**Figure 8 microorganisms-09-02012-f008:**
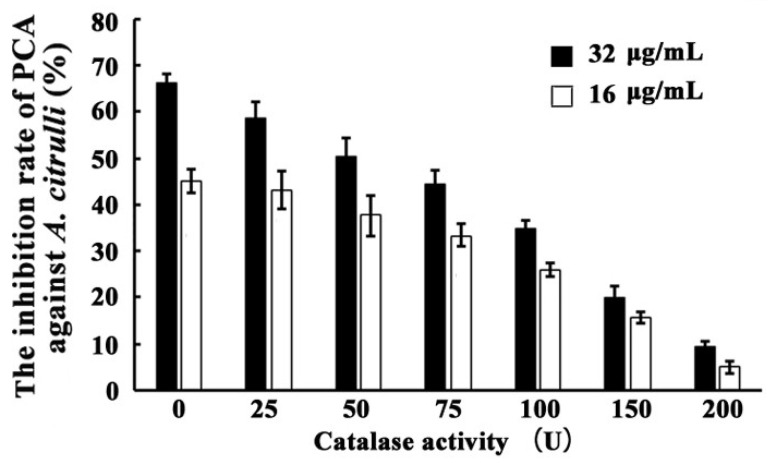
Inhibition effect of PCA against *A. citrulli* by adding exogenous CAT, 0, 25, 50, 75, or 100 U. The experiment was repeated three times.

## Data Availability

The genome data presented in this study are openly available in GenBank under the accession NZ_AWWJ00000000.1 (*P. chlororaphis* YL-1).

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
