# Peer review of "Phenazine-1-carboxylic Acid Produced by *Pseudomonas* *chlororaphis* YL-1 Is Effective against *Acidovorax citrulli"

_microorganisms, 2021, doi:10.3390/microorganisms9102012_

Round 1
Reviewer 1 Report
In this well designed work, the authors study three gene clusters (phenazine, pyrrolnitrin, pyoverdin) in P. chlororaphis YL-1 that may have an inhibitory effect in A. citrulli and first find that the main inhibition is via the phenazine cluster. Next, they perform gene expression analysis and try to elucidate the mechanism of action of phenazine. They observe that the inhibitory effect is by affecting genes that play a significant role in oxidative stress, which is in accordance with other studies in the field. Overall, the paper was well written (with some minor typos), the methods and results were well explained, and the manuscript was well structured.
My “major correction” which is not really major is that the authors need to show to what extent their findings are transferable to other P. chlororaphis strains. For this, they need to look at the supplementary files of Nikolaidis et al., 2020 (DOI: 10.3390/d12080289), where these authors performed comparative genomic and core genome analyses of the various Pseudomonas groups, including the P. chlororaphis group. In the supplementary excel file (sheet 4) of that study, Nikolaidis et al show the 3587 core genes of the P. chlororaphis group (that was based on 43 P. chlororaphis complete genomes) and also shows the presence of the P. chlororaphis core genes in other Pseudomonads. The authors need to find their genes of interest in that list and show somewhere in the results or at least in the supplementary material how many of their genes of interest are core genes in the P. chlororaphis group. Next, they need to discuss these findings with a few sentences or small paragraph in the Discussion.
It would be worth mentioning somewhere in the Introduction or Discussion that in NCBI genome, the YL1 strain is phylogenetically very close to the O6 strain that has also been studied for the production of antifungal compounds phenazine and pyrrolnitrin (see Park et al., 2011; PMID: 21362001)
The part of the introduction concerning pyoverdine and pyrrolnitrin could benefit from including the citations of Mossialos et al., 2007 (PMID: 17683275) and Pawar et al., 2019 (PMID: 31484394).
It may worth mentioning somewhere in the Discussion that PCA is categorized as an irritant in toxicology databases, such as comptox.
Line 19-20: Do the authors mean that PCA reduced the activity of Catalase in A. citrulli?
Line 32: under a favorable environment
Line 43: is to be sequentially
Line 55: needs a reference.
Line 62-66. Needs rephrasing.
Line 255: the authors mean p-value < 0.05
Line 443: needs a reference.
Line 525: to our knowledge
Line 516: Vibrio cholerae
Line 525: “to our knowledge”, also please rephrase
Author Response
In this well-designed work, the authors study three gene clusters (phenazine, pyrrolnitrin, pyoverdin) in P. chlororaphis YL-1 that may have an inhibitory effect in A. citrulli and first find that the main inhibition is via the phenazine cluster. Next, they perform gene expression analysis and try to elucidate the mechanism of action of phenazine. They observe that the inhibitory effect is by affecting genes that play a significant role in oxidative stress, which is in accordance with other studies in the field. Overall, the paper was well written (with some minor typos), the methods and results were well explained, and the manuscript was well structured.
Response: Thanks for the positive comments.
My “major correction” which is not really major is that the authors need to show to what extent their findings are transferable to other P. chlororaphis strains. For this, they need to look at the supplementary files of Nikolaidis et al., 2020 (DOI: 10.3390/d12080289), where these authors performed comparative genomic and core genome analyses of the various Pseudomonas groups, including the P. chlororaphis group. In the supplementary excel file (sheet 4) of that study, Nikolaidis et al show the 3587 core genes of the P. chlororaphis group (that was based on 43 P. chlororaphis complete genomes) and also shows the presence of the P. chlororaphis core genes in other Pseudomonads. The authors need to find their genes of interest in that list and show somewhere in the results or at least in the supplementary material how many of their genes of interest are core genes in the P. chlororaphis group. Next, they need to discuss these findings with a few sentences or small paragraph in the Discussion.
Response: Thanks for the valuable suggestions. This is an excellent research on genome analyses of the various Pseudomonas groups. According to the reports of Nikolaidis et al., 2020, phenazine biosynthetic gene phzF is present in 86% (37/43) of the analyzed P. chlororaphis strains while as a core gene existing in all tested P. aeruginosa strains (189/189). The pvdL gene was identified as a core gene in all tested P. chlororaphis strains (43/43) while 90% (171/189) in P. aeruginosa strains and 81% (212/262) in all the other Pseudomonas strains. These results indicate that both phenazine and pyoverdine genes distribute widely across the genomes of the genus Pseudomonas. We agree with the reviewer that genomewide comparison of strain YL-1 with those of P. chlororaphis other strains. However, the genomic analysis if out of the research focus of this manuscript. In addition, more research is needed to investigate any possible variations of PCA produced in other Pseudomonas species. The information has been added in the discussion of revised manuscript.
It would be worth mentioning somewhere in the Introduction or Discussion that in NCBI genome, the YL1 strain is phylogenetically very close to the O6 strain that has also been studied for the production of antifungal compounds phenazine and pyrrolnitrin (see Park et al., 2011; PMID: 21362001)
Response: Thanks for the suggestion. The information has been added in the discussion of revised manuscript.
The part of the introduction concerning pyoverdine and pyrrolnitrin could benefit from including the citations of Mossialos et al., 2007 (PMID: 17683275) and Pawar et al., 2019 (PMID: 31484394).
Response: Thanks for the suggestion. The information has been added in the introduction of revised manuscript.
It may worth mentioning somewhere in the Discussion that PCA is categorized as an irritant in toxicology databases, such as comptox.
Response: Thanks for the suggestion. The information has been added in the revised manuscript.
Line 19-20: Do the authors mean that PCA reduced the activity of Catalase in A. citrulli?
Response: We apologize for the misunderstanding caused by our unclear description. It has been corrected in the revised manuscript. ------Gene expression analyses revealed that PCA enhanced the accumulation of reactive oxygen species (ROS) and increased the activity of catalase (CAT) in A. citrulli. The inhibition effect of PCA against A. citrulli was lowered by adding exogenous CAT.
Line 32: under a favorable environment
Response: Thanks for the suggestion. It has been corrected in the revised manuscript.
Line 43: is to be sequentially
Response: Thanks for the suggestion. It has been corrected in the revised manuscript.
Line 55: needs a reference.
Response: Thanks for the suggestion. It has been corrected in the revised manuscript.
Line 62-66. Needs rephrasing.
Response: Thanks for the suggestion. It has been corrected in the revised manuscript.
Line 255: the authors mean p-value < 0.05
Response: That’s right. Thanks for the suggestion. It has been corrected in the revised manuscript.
Line 443: needs a reference.
Response: Thanks for the suggestion. It has been corrected in the revised manuscript.
Line 525: to our knowledge
Response: Thanks for the suggestion. It has been corrected in the revised manuscript.
Line 516: Vibrio cholerae
Response: We apologize for the carelessness. It has been corrected in the revised manuscript.
Line 525: “to our knowledge”, also please rephrase
Response: Same suggestion. Please see above.

Reviewer 2 Report
This manuscript shows the effect of PCA produced by Pseudomonas chlororaphis YL-1 against Acidovorax citrulli. The authors have created several mutants and studied the compound inhibiting A. citrulli. Although the study seems interesting, there are few issues that must be addressed for a broad readership. I have three major concerns: - While saying that PCA is ‘effective’ against A. citrulli, there must be some reference (probably an antibiotic) for comparison and clarity. At a dose of 32 µg/mL, there was only about 67% inhibition of A. citrulli (NOT much effective indeed. A higher dose may perform better, but was not included in this study!, will existing arsenal of antibiotics will exert better results at lower doses? Could have been discussed). - The author suggests that “performance of phenazine producers used as biological control agents for soil-borne plant pathogens”. This is quite intriguing as the author also mentioned that “Phenazine production plays role in survival and persistence of Pseudomonas spp. (some of which are plant pathogens as well!!) in the rhizosphere and in many physiological processes, including biofilm formation and iron reduction”. How can this compound (PCA) find its use as a biocontrol agent? - Figure 7: Why acetone alone has a similar influence as that of PCA (16 and 32 µg/mL) for both CAT activity and SOD? Other observations: - The MS spectra shown here is very different compared to that of the https://pubchem.ncbi.nlm.nih.gov/compound/Phenazine-1-carboxylic-acid#section=Mass-Spectrometry In addition, the molecular weight of the compound is 224 [but m/z is a bit higher, maybe discussed]. - Figure 6 A to 6C: The X-axis labels should be the same for better comparison.Author Response
Please see attached file. Thanks.

Reviewer 3 Report
The present manuscipt descibes the role and action of phenazine-1 carboxylic acid as an antibacterial factor of P. chlororaphis YL-1 against Acidovorax citrulli. In general this study is well designed and conducted but authors should address the following issues before this work is published in Microorganisms:
__ It is rather confusing for unfamiliar readers the part regarding the mutants used in this study. Authors should describe the role of impaired genes and provide appropriate citations. For example hcnB gene is essential for cyanide production , a known antimicrobial factor against bacteria, oomycetes etc (Laville et al 1998 J. Bacteriol. ) pvdL encodes a non-ribosomal peptide synthetase implicated in pyoverdine chromophore biosynthesis (Mossialos et al. 2002 Mol. Microbiol) etc
__It is known that LB medium is iron-rich. Therefore I do not understand why authors tested the antibacterial activity of pvdL mutant against A. citrulli on LB plates. It was expected that pyoverdine is not produced therefore its role could not be elucidated under these growth conditions. It would be more appropriate to test thuis mutant in iron-depleted medium or even better to construct a double mutant impaired in pyoverdine-PCA production and tested in comparison to single pvdL mutant and wilt type strain under iron limitation. I think this experiment would reveal interesting results.
Author Response
Please see the atatched file. Thanks.

Round 2
Reviewer 2 Report
The authors have clarified the issues raised previously. However, by the rebuttal, it is clear that the present manuscript lacks novelty: PCA is already known potential biocontrol agent produced by several Pseudomonas species, works against A. citrulii, and affects ROS (previous literature available in the public domain from the year 2016 onwards). In addition, the similar effect of solvent (acetone) on CAT and SOD makes the data confusing as to whether it was solely due to PCA or the solvent effect?
Based on the above observation, I vote against the publication of this manuscript.
Author Response
The authors have clarified the issues raised previously. However, by the rebuttal, it is clear that the present manuscript lacks novelty: PCA is already known potential biocontrol agent produced by several Pseudomonas species, works against A. citrulii, and affects ROS (previous literature available in the public domain from the year 2016 onwards).
Response: Thanks for the suggestions. In the new revised manuscript, we have summarized the novelty of the research including three points below:
- Previous results showed that the main antibacterial secondary metabolite produced by strain YL-1 is pyoverdine in low-iron condition. Strain YL-1 and PVD-deficient mutants secreted different antibacterial compounds when cultured on LB plates as compared with low-iron conditions [39]. In this study, we found that main antibacterial compound produced by strain YL-1 on LB plates was PCA. These findings of the research further demonstrate that chlororaphis YL-1 can produce multiple antimicrobial compounds to target various microbial organisms under different conditions.
- Although one paper reported that PCA can inhibit the growth of citrulli in vitro [51], however, the mechanism of PCA against A. citrulli remain unclear. In this study, A. citrulli can hardly grow on LB plates and the inhibitory efficiency in liquid co-culture test was 66.11% when treated by 32 μg/mL of PCA. The EC50 value of PCA against A. citrulli was 19.81 μg/mL. Moreover, we also found that PCA enhanced the accumulation of ROS and increased the activity of CAT in A. citrulli. The inhibition of PCA against A. citrulli was reduced by adding exogenous CAT (Fig. 8).
- Our findings demonstrate that different bacteria have various CATs that are encoded and activated by different genes although the accumulation of ROS in citrulli affected by PCA is similar with those in Xoo [49]. There is evidence that expressionintensity of these different genes determines the difference of antioxidant capacity of bacteria. For example, it was reported that Xooc has a stronger antioxidant system than Xoo and accumulates lower of ROS in response to redox compounds by deleting or exchanging the antioxidant-related genes, although Xoo and Xooc are very closely related [49]. Therefore, our results provide a scientific basis for the study of antioxidant capacity, virulence and control of A. citrulli in the future.
In addition, the similar effect of solvent (acetone) on CAT and SOD makes the data confusing as to whether it was solely due to PCA or the solvent effect?
Response: Thanks for the suggestion. We apologize for the mistake (wrong picture number and related data) in last revised manuscript. It has been corrected in this new vision. ----- It should be noted that acetone (the solvent of PCA) reaching a certain concentration would have inhibitory effect on A. citrulli (data not shown); therefore, the concentration of acetone in the research was set as 0.4% (v/v). A. citrulli was not inhibited and can grow well on LB plates (Fig. 4) and in LB liquid medium (Fig. 5) containing 0.4% acetone (v/v). Although 0.4% acetone has a slight influence on CAT activity in A. citrulli (Fig. 7), it was showed significant difference as compared with 32 μg/mL PCA from 6 h to 10 h. No significant difference was found on SOD activity 6 h after treated by acetone, 16 or 32 μg/mL PCA.
Reviewer 3 Report
Indeed, authors significantly improved this manuscript which might now be accepted for publication in Microorganisms
Author Response
Indeed, authors significantly improved this manuscript which might now be accepted for publication in Microorganisms
Response: Thank for your comment and recommendation.